# Exploiting Data Sparsity in Secure Cross-Platform Social Recommendation

**Jamie Cui**[1], **Chaochao Chen**[2,1*], **Lingjuan Lyu**[3], **Carl Yang**[4], and **Li Wang**[1]

[1]Ant Group
[2]Zhejiang University
[3]Sony AI
[4]Emory University
[*]Corresponding author, email: `zjuccc@zju.edu.cn`

## Abstract

Social recommendation has shown promising improvements over traditional systems since it leverages social correlation data as an additional input. Most existing works assume that all data are available to the recommendation platform. However, in practice, user-item interaction data (e.g., rating) and user-user social data are usually generated by different platforms, both of which contain sensitive information. Therefore, *How to perform secure and efficient social recommendation across different platforms, where the data are highly-sparse in nature* remains an important challenge. In this work, we bring secure computation techniques into social recommendation, and propose $S^3$Rec, a sparsity-aware secure cross-platform social recommendation framework. As a result, $S^3$Rec can not only improve the recommendation performance of the rating platform by incorporating the sparse social data on the social platform, but also protect data privacy of both platforms. Moreover, to further improve model training efficiency, we propose two secure sparse matrix multiplication protocols based on homomorphic encryption and private information retrieval. Our experiments on two benchmark datasets demonstrate that $S^3$Rec improves the computation time and communication size of the state-of-the-art model by about $40\times$ and $423\times$ in average, respectively.

## 1 Introduction

The recent advances of social recommendation have achieved remarkable performances in recommendation tasks [12, 28]. Unlike traditional methods, social recommendation leverages user-item rating data (e.g. from Netflix) with user-user social data (e.g. from Facebook) to facilitate model training. The intuition behind this setup is that Facebook's social data is much better than Netflix's social data in both quantity and quality, and those social data at Facebook can help to improve Netflix's recommendation performance. However, the cross-platform nature, the high sparsity and sensitivity of recommendation/social data make social recommendation hard-to-deploy in the real world [5]. In summary, the main problem we are facing is,

*How to perform **secure** and **efficient** social recommendation across different platforms, where the data are **highly-sparse** in nature?*

Specifically, we focus on the problem of collaborative social recommendation in the two-party model, where one party (denoted as $P_0$) is a rating platform that holds user-item rating data, and the other party (denoted as $P_1$) is a social platform that holds user-user social data. We also assume that the adversaries are semi-honest, which is commonly used in the secure computation literature [9]. That

35th Conference on Neural Information Processing Systems (NeurIPS 2021).

is to say, the adversary will not deviate from the pre-defined protocol, but will try to learn as much information as possible from its received messages.

**Choices of privacy enhancing techniques.** Currently, many anonymization techniques have been used in publishing recommendation data, such as *k-anonymity* and *differential privacy* [11]. On the other hand, cryptographic methods like *secure multiparty computation* (MPC) [11] and *homomorphic encryption* (HE) have been proposed to enable calculation on the protected data. Since k-anonymity has been demonstrated risky in practice (e.g., the re-identification attack on Netflix Prize dataset [22]), and differential privacy introduces random noises to the dataset which eventually affects model accuracy [10, 30], we consider they are not the ideal choice for our framework. Instead, we choose a combination of cryptographic tools (i.e., MPC and HE, but mainly MPC) which allows multiple parties to jointly compute a function depending on their private inputs while providing security guarantees.

**Choices of social recommendation model.** In literature, many social recommendation models have been proposed [8, 18, 27] using matrix factorization or neural networks. Existing MPC-based neural network protocols [21, 29] usually suffer from accuracy loss and inefficiency due to their approximation of non-linear operations. Especially for the case of social recommendation, training data could exceed to millions, and this makes NN-based model a less ideal choice. Therefore, we choose the classic social recommendation model, Soreg [18], as a typical example, and present how to build a secure and efficient version of Soreg under cross-domain social recommendation scenario.

**Dealing with sparse data in secure machine learning.** One important property of social recommendation data is its high sparsity. Take LibraryThing dataset [32] for example, its social matrix density is less than 0.02%. Recently, Schoppmann et al. introduced the ROOM framework [26] for secure computation over sparse data. However, their solution only works on column-sparse or row-sparse data, and in addition, it requires secure matrix multiplication protocol (for instance, based on Beaver's multiplication triple). Chen et al. proposed a secure protocol for a sparse matrix multiplies a dense matrix [6], which combines homomorphic encryption and secret sharing, but it only works well when the dense matrix is small. Different from their work, in this paper, we propose a PIR-based matrix multiplication which does not reply on pre-generated correlated randomness.

**Our framework.** In this paper, we propose $S^3$Rec, a sparsity-aware secure cross-platform social recommendation framework. Starting with the classic Soreg model, we observe that the training process of Soreg involves two types of calculation terms: (1) the *rating term* which could be calculated by $P_0$ locally, and (2) the *social term* which needs to be calculated by $P_0$ and $P_1$ collaboratively. Therefore, the key to $S^3$Rec is designing secure and efficient protocols for calculating the social term.

To begin with, we first let both parties perform local calculation. Then both parties invoke a secure social term calculation protocol and let $P_0$ finally receive the plaintext social term, and update the model accordingly. In this way, the security of our protocol relies significantly on the secure social term calculation protocol (for simplicity, we refer this protocol as the 'ST-MPC' protocol), and we propose a secure instantiation and prove its security. Similarly, the efficiency of $S^3$Rec relies heavily on the performance of ST-MPC, and at the core, it relies on the efficiency of a matrix multiplication protocol. The naïve secure matrix multiplication protocol is traditionally evaluated through Beaver's triples [3], and has $O(km^2)$ asymptotic communication complexity, where $k$ is the dimension of latent factors and $m$ is the number of users. To improve the communication efficiency, we propose two secure sparse matrix multiplication protocols for ST-MPC, based on two sparsity settings: (1) *insensitive sparsity*, which is a weaker variant of matrix multiplication where we assume both parties know the locations of non-zero values in the sparse matrix, and (2) *sensitive sparsity*, which is also a weaker variant of matrix multiplication, but stronger than (1), and we assume 'only' the number of zeros is public. Nevertheless, we present secure constructions for MatrixMul in both cases by leveraging two cryptography primitives called *Private Information Retrieval* (PIR) [1] and *Homomorphic Encryption* (HE) [24]. PIR can hide the locations of the non-zero values in the sparse matrix while HE enables additions and multiplications on ciphertexts. To this end, we drop the communication complexity of secure MatrixMul to $O(km)$ for the insensitive sparsity case and to $O(\alpha km)$ for the sensitive sparsity case, where $\alpha$ denotes the density of user social matrix.

**Summary of our experimental results.** We conduct experiments on two popularly used dataset, i.e., Epinions [19] and LibraryThing [32]. The results demonstrate that (1) $S^3$Rec achieves the same

performance as existing social recommendation models, and (2) $S^3$Rec improves the computation time and communication size of the state-of-the-art (SeSoRec) by about $40\times$ and $423\times$ in average.

**Contributions.** We summarize our main contributions below: (1) We propose $S^3$Rec, a privacy-preserving cross-platform social recommendation framework, which relies on a general protocol for calculating the social term securely; (2) We propose two secure sparse matrix multiplication protocols based on different sparsity visibility, i.e., insensitive sparsity and sensitive sparsity. We prove that both protocols are secure under semi-honest adversaries; and (3) We empirically evaluate the performance of $S^3$Rec on benchmark datasets.

## 2  Tools and Recommendation Model

**Notation.** We use $[n]$ to denote the set $\{1, ..., n\}$, and $|x|$ to denote the bit length of $x$. In terms of MPC, we denote a secret shared value of $x$ in $\mathbb{Z}_N$ as $[\![x]\!]$, where $N$ is a positive integer. Also, we let $[\![x]\!]_0$ denote $P_0$'s share, and $[\![x]\!]_1$ denote $P_1$'s share, where $[\![x]\!] = [\![x]\!]_0 + [\![x]\!]_1 \in \mathbb{Z}_N$. We also use $\leftarrow$ to denote the assignment of variables, e.g., $x \leftarrow 4$.

### 2.1  Tools

In this section, we introduce several secure computation tools used in our work.

**Multi-Party Computation (MPC).** MPC is a cryptographic tool which enables multiple parties (say, $n$ parties) to jointly compute a function $f(x_1, ..., x_n)$, where $x_i$ is $i$-th party's private input. MPC protocols ensure that, at the end of the protocol, parties eventually learn nothing but their own input and the function output. MPC has been widely-used in secure machine learning systems such as PrivColl [31] and CrypT-Flow [15], most of which support a wide range of linear (e.g. addition, multiplication) and non-linear functions (e.g. equality test, comparison). Here, we present three popular MPC protocols (addition, multiplication, and matrix multiplication), which we will use later in our protocol,

---

MatrixMul($\mathbf{X}, \mathbf{Y}$)

     (**Offline**) Generate $km^2$ Beaver's triples
1 :   $\forall x_{i,j} \in \mathbf{X}$, $P_0$ invokes $[\![x_{i,j}]\!] \leftarrow \mathsf{Shr}(x_{i,j})$
2 :   $\forall y_{i,j} \in \mathbf{Y}$, $P_1$ invokes $[\![y_{i,j}]\!] \leftarrow \mathsf{Shr}(y_{i,j})$
3 :   **foreach** $i \in [k], j \in [m]$, let $[\![z_{i,j}]\!] = 0$,
4 :     **foreach** $a \in [m], b \in [m]$,
5 :       $[\![\mathsf{tmp}]\!] \leftarrow \mathsf{Mul}([\![x_{i,a}]\!], [\![y_{b,j}]\!])$
6 :       $[\![z_{i,j}]\!] \leftarrow \mathsf{Add}([\![\mathsf{tmp}]\!], [\![z_{i,j}]\!])$
7 :     **endfor**
8 :   **endfor**
9 :   **return** $[\![\mathbf{Z}]\!]$

Figure 1: Secure matrix multiplication protocol, where Shr is a secret sharing algorithm.

---

$\mathsf{Add}([\![x]\!], [\![y]\!])$: Take two shares as inputs from both parties, $P_{b \in \{0,1\}}$ locally calculate and return $[\![x]\!]_b + [\![y]\!]_b$.

$\mathsf{Mul}([\![x]\!], [\![y]\!])$: Take two shares as inputs from both parties, then evaluate using Beaver's Triples [3].

**Homomorphic Encryption (HE) scheme.** HE is essentially a specific type of encryption scheme which allows manipulation on encrypted data. More specifically, HE involves a key pair $(\mathsf{pk}, \mathsf{sk})$, where the public key $\mathsf{pk}$ is used for encryption and the secret key $\mathsf{sk}$ is used for decryption. In this work, we use an additive HE scheme (i.e., Paillier [24]) which allows the following operations:

$\mathsf{Enc_{pk}}(x) \oplus \mathsf{Enc_{pk}}(y)$: addition between two ciphertexts, returns $z = \mathsf{Enc_{pk}}(x + y)$;

$\mathsf{Enc_{pk}}(x) \otimes y$: multiplication between a ciphertext and a plaintext, returns $z = \mathsf{Enc_{pk}}(x \cdot y)$.

**Private Information Retrieval (PIR).** Now, we introduce single-server PIR [1]. In this setting, we assume there is a server and a client, where the server holds a database $\mathsf{DB} = \{d_1, ..., d_n\}$ with $n$ elements, and the client wants to retrieve $\mathsf{DB}_i$ while hiding the query index $i$ from the server. Roughly, a PIR protocol consists of a tuple of algorithm $(\mathsf{PIR.Query}, \mathsf{PIR.Response}, \mathsf{PIR.Extract})$. First, the client generates a query $q \leftarrow \mathsf{PIR.Query}(i)$ from an index $i$, and then sends query $q$ to the server. The server then is able to generate a response $r \leftarrow \mathsf{PIR.Response}(\mathsf{DB}, q)$ based on the query and

database DB, and returns $r$ to the client. Finally, the client extracts the result from server's response $\text{DB}_i \leftarrow \text{PIR.Extract}(r)$.

| Client | Server |
|---|---|
| $q \leftarrow \text{PIR.Query}(i)$ $\quad\xrightarrow{\quad q \quad}$ | |
| $\xleftarrow{\quad r \quad}$ $\quad r \leftarrow \text{PIR.Response}(\text{DB}, q)$ | |
| $\text{DB}_i \leftarrow \text{PIR.Extract}(r)$ | |

Figure 2: An overview of Private Information Retrieval (PIR).

## 2.2 Recommendation model

Recall that we assume there are two platforms, a rating platform $P_0$, and a social platform $P_1$. We assume $P_0$ holds a private rating matrix $\mathbf{R} \in \mathbb{R}^{m \times n}$, and $P_1$ holds a private user social matrix $\mathbf{S} \in \mathbb{R}^{m \times m}$, where $n$ and $m$ denote the number of items and their common users, respectively. Also, we denote the user latent factor matrix as $\mathbf{U} \in \mathbb{R}^{k \times m}$ and item latent factor matrix as $\mathbf{V} \in \mathbb{R}^{k \times n}$, where $k$ is the dimension of latent factors. We further define an indication matrix $\mathbf{I} \in \mathbb{R}^{m \times n}$, where $I_{i,j}$ denotes whether user $i$ has rated item $j$.

Existing work [27] summarizes factorization based social recommendation models as the combination of a "basic factorization model" and a "social information model". To date, different kinds of social information models have been proposed [18, 14], and their common intuition is that users with social relations tend to have similar preferences. In this work, we focus on the classic social recommendation model, i.e., Soreg [18], which aims to learn $\mathbf{U}$ and $\mathbf{V}$ by minimizing the following objective function,

$$\sum_{i=1}^{m} \sum_{j=1}^{n} \frac{1}{2} I_{i,j} \left( r_{i,j} - \mathbf{u}_{*,i}^T \mathbf{v}_{*,j} \right)^2 + \frac{\lambda}{2} \sum_{i=1}^{m} \|\mathbf{u}_{*,i}\|_F^2 + \frac{\lambda}{2} \sum_{j=1}^{n} \|\mathbf{v}_{*,j}\|_F^2 + \frac{\gamma}{2} \sum_{i=1}^{m} \sum_{f=1}^{m} s_{i,f} \|\mathbf{u}_{*,i} - \mathbf{u}_{*,f}\|_F^2, \tag{1}$$

where the first term is the basic factorization model, the last term is the social information model, and the middle two terms are regularizers, $\|\cdot\|_F^2$ is the Frobenius norm, $\lambda$ and $\gamma$ are hyper-parameters. If we denote $\mathbf{D} \in \mathbb{R}^{m \times m}$ as a diagonal matrix with diagonal element $d_b = \sum_{c=1}^{m} s_{b,c}$ and $\mathbf{E} \in \mathbb{R}^{m \times m}$ as a diagonal matrix with diagonal element $e_i = \sum_{b=1}^{m} s_{b,i}$. The gradients of $\mathcal{L}$ in Eq. (1) with respect to $\mathbf{U}$ and $\mathbf{V}$ are,

$$\frac{\partial \mathcal{L}}{\partial \mathbf{U}} = \underbrace{-\mathbf{V} \left( \left( \mathbf{R} - \mathbf{U}^T \mathbf{V} \right)^T \circ \mathbf{I} \right) + \lambda \mathbf{U}}_{\text{Rating term: computed by } P_0 \text{ locally}} + \underbrace{\frac{\gamma}{2} \mathbf{U}(\mathbf{D}^T + \mathbf{E}^T) - \gamma \mathbf{U} \mathbf{S}^T}_{\text{Social term: computed by } P_0 \text{ and } P_1 \text{ collaboratively}}, \tag{2}$$

$$\frac{\partial \mathcal{L}}{\partial \mathbf{V}} = \underbrace{-\mathbf{U} \left( \left( \mathbf{R} - \mathbf{U}^T \mathbf{V} \right)^T \circ \mathbf{I} \right) + \lambda \mathbf{V}}_{\text{Rating term: computed by } P_0 \text{ locally}} . \tag{3}$$

# 3 Framework

We summarize our proposed S$^3$Rec framework in Figure 3. To begin with, we assume that party $P_0$ holds the rating matrix $\mathbf{R}$ and $P_1$ holds the social matrix $\mathbf{S}$. At first, $P_0$ randomly initializes $\mathbf{U} \leftarrow_\$ \mathbb{R}^{k \times m}$ and $\mathbf{V} \leftarrow_\$ \mathbb{R}^{k \times n}$. Then, for each iteration (while the model dose not coverage), we let $P_0$ and $P_1$ jointly evaluate the social term defined in Eq 2. $P_0$ then locally calculates the rating term in Eq 2 and Eq 3, as well as $\partial \mathcal{L}/\partial \mathbf{U}$ and $\partial \mathcal{L}/\partial \mathbf{V}$. Party $P_0$ then locally updates $\mathbf{U}$ and $\mathbf{V}$ accordingly and ends the iteration.

**Communication efficiency.** In our framework, the only communication between two parties occurs in the ST-MPC protocol. Since we choose additive secret sharing, the Add protocol contains only local computation, we claim that the communication efficiency of S$^3$Rec significantly relies on the efficiency of matrix multiplication protocol. We give a popular MatrixMul protocol in Figure

**Global Parameter**: Regularization strength $\gamma$, and learning rate $\theta$.

**Input:** Private rating matrix $\mathbf{R}$ from platform $P_0$, private user social matrix $\mathbf{S}$ from platform $P_1$.

**Output**: Platform $P_0$ receives the user latent matrix $\mathbf{U}$ and item latent matrix $\mathbf{V}$.

1 :     Platform $P_0$ initializes $\mathbf{U}$ and $\mathbf{V}$,

2 :     **while** not coverage,

3 :        $P_0$ and $P_1$ securely calculate the social term $\leftarrow$

4 :        $P_0$ locally computes the rating terms

5 :        $P_0$ locally updates $\mathbf{U}$ by $\mathbf{U} \leftarrow \mathbf{U} - \theta \cdot \partial\mathcal{L}/\partial\mathbf{U}$

6 :        $P_0$ locally updates $\mathbf{V}$ by $\mathbf{V} \leftarrow \mathbf{V} - \theta \cdot \partial\mathcal{L}/\partial\mathbf{V}$

7 :     **endwhile**

8 :     **return** $\mathbf{U}$ and $\mathbf{V}$ to platform $P_0$

---

ST-MPC$(\gamma, \mathbf{U}, \mathbf{D}, \mathbf{E}, \mathbf{S})$

---

1 :    $[\![\mathbf{R}_0]\!] \leftarrow$ MatrixMul$(\gamma\mathbf{U}/2, \mathbf{D}^T + \mathbf{E}^T)$

2 :    $[\![\mathbf{R}_1]\!] \leftarrow$ MatrixMul$(-\gamma\mathbf{U}, \mathbf{S}^T)$

3 :    $[\![\mathbf{R}]\!] \leftarrow$ Add$([\![\mathbf{R}_0]\!], [\![\mathbf{R}_1]\!])$

4 :    **return** Rec$([\![\mathbf{R}]\!])$ to $P_0$

Figure 3: Our proposed S$^3$Rec framework, where MatrixMul stands for secure matrix multiplication protocol, Add stands for secure add protocol, Rec stands for reconstruction protocol for secret sharing.

1 and analyze its efficiency in our framework. The protocol in Figure 1 requires $km^2 \log_2 N$ bit online communication, where $m$ is the number of users and $k$ is the dimension of latent factors. As for the usual case where the number of users is $\approx 10^4$, $k = 10$, and $\log_N = 64$, one invocation of MatrixMul protocol would have a total communication of around 7.4GB. Considering 100 iterations of our framework, this leads to $\approx 1491$GB communication, which is impractical. Fortunately, the social matrices ($\mathbf{D}$, $\mathbf{E}$, and $\mathbf{S}$) are highly sparse in social recommendation. In the following section, we propose a PIR-based sparse matrix multiplication protocol with better communication efficiency.

## 3.1 Secure sparse matrix multiplication

Essentially, any matrix could be represented by a value vector and a location vector, where the value vector contains all non-zero values and the location vector contains locations of those values. That is, a sparse matrix $\mathbf{Y} \in \mathbb{R}^{m \times m}$ can be represented by a pair of vectors $(l_y \in \mathbb{N}_{m^2}^t, v_y \in \mathbb{R}^t)$, where $t$ is the number of non-zero values in $\mathbf{Y}$.

**Dense-sparse matrix multiplication.** Considering the case where $\mathbf{X} \in \mathbb{R}^{k \times m}$ is the dense matrix from $P_0$ and $\mathbf{Y} \in \mathbb{R}^{m \times m}$ is the sparse matrix from $P_1$. Now we consider the following two cases.

*Case 1: insensitive sparsity, i.e., insensitive $l_y$ and sensitive $v_y$.* This refers to the case where the locations of zero values are public or contain no sensitive information. Take the social matrices ($\mathbf{D}$ and $\mathbf{E}$) for example, both of them are diagonal, and thus the location vector is insensitive while the value vector is still sensitive.

Our protocol mainly works as follows. First, $P_0$ and $P_1$ parse $\mathbf{X}$ and $\mathbf{Y}$ into two tables $T_x$ and $T_y$ separately, where the value set of each bin in $T_x$ is a subset of one row in $\mathbf{X}$, that is, $T_x(i) \subseteq x_{i,*}$. Similarly, bin set in $T_y$ is a subset of one column in $\mathbf{Y}$, $T_y(i) \subseteq y_{*,i}$. The intuition behind is to use bins to contain only

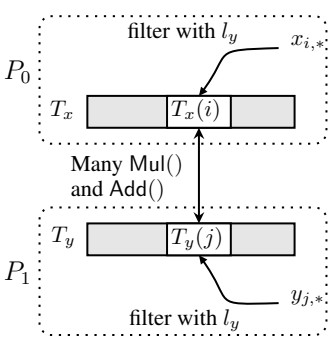

Figure 4: Matrix multiplication with insensitive sparsity.

the necessary values needed to calculate the output value (which means filter out the zero multiplies in each bin). Take the first bin for example (that is, $T_x(0)$ and $T_y(0)$), for $j \in [m]$, $T_x(0)$ contains all $x_{0,j}$ where $y_{j,0}$ is a non-zero value, and $T_y(0)$ contains all non-zero $y_{j,0}$. In order to get the final result, we perform the secure inner product protocol on $T_x(0)$ and $T_y(0)$, and denote the result as $[\![z_{0,0}]\!]$. We show the high level idea in Figure 4. By doing this, our protocol concretely consumes $k|l_y|$ Beaver's triples and therefore has $O(k|l_y|)$ online communication complexity. Figure 5 shows the technical details of our proposed protocol for case 1. For Line 1 in ST-MPC (Figure 3), clearly both parties know that $\mathbf{D}$ and $\mathbf{E}$ are diagonal matrices, that is, $|l_y| = m$. Therefore, our proposed protocol in Figure 4 can drop the complexity from $O(km^2)$ to $O(km)$.

---

**MatrixMul($\mathbf{X}, \mathbf{Y}$) with insensitive sparsity**

      (**Offline**) Generate $km^2$ Beaver's triples

1 :   $\forall (i,j) \in l_y$, $P_1$ pushes $y_{i,j}$ into $T_y(j)$

2 :   **for** $j \in [k]$ **do**

3 :      $P_1$ lets $T_y = \emptyset$

4 :      $\forall a \in [m], b \in [m]$, if $(i,a) \in l_y$, $P_0$ pushes $x_{a,b}$ into $T_x(a)$

5 :      **for** $j \in [m]$ **do**

6 :         Both parties let $[\![z_{i,j}]\!] = 0$, then, for all values $v \in T_x(i), u \in T_y(j)$

7 :         $P_0$ invokes $[\![v]\!] \leftarrow \mathsf{Shr}(v)$, $P_1$ invokes $[\![v]\!] \leftarrow \mathsf{Shr}(u)$

8 :         $[\![z_{i,j}]\!] = \mathsf{Add}(\mathsf{Mul}([\![v]\!], [\![u]\!]), [\![z_{i,j}]\!])$

9 :      **endfor**

10 :  **endfor**

---

**MatrixMul($\mathbf{X}, \mathbf{Y}$) with sensitive sparsity**

      (**Offline**) $P_0$ generates an additive HE key pair $(\mathsf{pk}, \mathsf{sk})$, then sends $\mathsf{pk}$ to $P_1$

1 :   $\forall i \in [k], j \in [m]$, $P_0$ lets $e_{i,j} = \mathsf{Enc}(\mathsf{pk}, x_{i,j})$, and lets $\mathbf{E}$ be the encrypted matrix

2 :   $\forall (i,j) \in l_y$, $P_1$ pushes $y_{i,j}$ into $T_y(j)$, also, $P_1$ invokes $q_{i,j} \leftarrow \mathsf{PIR.Query}(i + jk)$

3 :   $P_1$ sends the query set (denoted as $\mathbf{q}$) to $P_0$

4 :   $\forall q_{i,j} \in \mathbf{q}$, $P_0$ invokes $r_{i,j} \leftarrow \mathsf{PIR.Response}(\mathbf{E}, q_{i,j})$.

5 :   $P_0$ sends the response set (denoted as $\mathbf{r}$) to $P_1$

6 :   $\forall r_{i,j} \in \mathbf{r}$, $P_1$ invokes $e_{i,j} \leftarrow \mathsf{PIR.Extract}(r_{i,j})$ and pushes $e_{i,j}$ to $T'_e(i)$

7 :   **for** $i \in [k], j \in [m]$ **do**

8 :      $P_0$ lets $\beta_{i,j} = \mathsf{Enc}_{\mathsf{pk}}(0)$

9 :      $\forall v \in T'_e(i), u \in T_y(j)$, $P_0$ invokes $\beta_{i,j} = v \otimes u \oplus \beta_{i,j}$

10 :     $P_0$ samples random numbers $g_{i,j} \leftarrow\!\!\$\, \mathbb{Z}_\delta$, then lets $\beta_{i,j} = g_{i,j} \oplus \beta_{i,j}$

11 :     $P_0$ sends $\beta_{i,j}$ to $P_1$, then lets $[\![z_{i,j}]\!]_0 = -g_{i,j}$

12 :  **endfor**

13 :  $P_0$ decrypts all receiving messages and lets $[\![z_{i,j}]\!]_1 = \mathsf{Dec}_{\mathsf{sk}}(\beta_{i,j})$

14 :  **return** $[\![Z]\!]$

---

Figure 5: Dense-sparse MatrixMul($\mathbf{X}, \mathbf{Y}$) with insensitive and sensitive sparsity protocols, where we have $\mathbf{X} \in \mathbb{R}^{k \times m}, \mathbf{Y} \in \mathbb{R}^{m \times m}$.

**Lemma 1.** *The first protocol in Figure 5 is secure against semi-honest adversary if we assume the existence of secure addition and multiplication semi-honest MPC protocols.*

*Proof.* Please find the proof in the Technical Appendix.     □

*Case 2: sensitive sparsity, i.e., sensitive $l_y$ and sensitive $v_y$.* For a more general case, where both the location vector and the value vector contain sensitive information. Take the social matrix $\mathbf{S}$ for instance, its location vector indicates the existence of a social relation between two users, its value vector further shows the strength of their relation, and both of which are sensitive.

In this case, both the dense matrix $\mathbf{X}$ and the entire sparse matrix $\mathbf{Y}$ are sensitive. Following the idea in case 1, the matrix multiplication protocol should first generate $T_x, T_y$ according to $v_x, v_y$ and $l_y$, and then perform the inner product multiplication for each aligned bins in $T_x, T_y$. Still, $P_1$ can generate $T_y$ according to its own inputs $v_y, l_y$. However, $P_0$ cannot generate $T_x$ directly, since $v_x$ is kept by itself while $l_y$ is held by $P_1$. We make a communication and computation trade-off by leveraging PIR techniques, and as a result, our PIR-based approach has lower concrete communication, and overall is faster than the baseline protocol.

We show the high-level idea of our PIR-based protocol in Figure 6. The intuition behind is to let $P_1$ obliviously filter each bin in $T_x$ since both value vector and location vector are sensitive. In summary,

first $P_0$ encrypts all the values in $T_x$, the encrypted table is denoted as $T_e$. Then $P_1$ and $P_0$ invoke PIR protocol, where $P_0$ acts as server and sets $T_e$ as PIR database, $P_1$ acts as client and parses $l_y$ to many PIR queries. At the end of PIR protocol, $P_1$ receives the encrypted and filtered table $T'_e$. Afterwards $P_1$ performs secure inner product evaluation. By doing this, the communication complexity drops from $O(km^2)$ to $O(\alpha km)$, compared with the simple solution. The details of our protocol are shown in Figure 5. For Line 2 in ST-MPC (Figure 3), the social matrix ($\mathbf{S}$) is sparse in nature, and thus our proposed protocol in Figure 6 can significantly improve its efficiency. In summary, with our proposed two secure MatrixMul protocols, one can securely calculate the social term efficiently. For instance, again considering the social recommendation with $\approx 10^4$ users, our proposal only requires a total of $\approx 3.6$GB communication for each iteration.

**Lemma 2.** *The second protocol in Figure 5 is secure against semi-honest adversary with the leakage of $|l_y|$ if we assume the existence of a secure PIR protocol.*

*Proof.* Please find the proof in the Technical Appendix. □

### 3.2 Security discussions of the social term

In S³Rec, two parties jointly calculate the social term $\gamma \mathbf{U}(\mathbf{D}^T + \mathbf{E}^T)/2 - \gamma \mathbf{US}^T$ and then reveal the social term to $P_0$ (see Eq. (2)). The security of S³Rec relies on whether $P_0$ can resolve the social matrix $\mathbf{S}^T$ given its own inputs $\mathbf{U}$ and the social term. We claim that this is difficult because, the number of equations $T$ (#epoch, 100 in our experiments) is much smaller than that of the variables $n$ (#user, much more than 100 in practice), which indicates that there are infinite solutions for this. In practice, $T < n$ can be easily satisfied for both the social platform and the rating platform. The reasons are two-folds. First, our proposed framework is secure against a semi-honest adversary (which is a popular threat model in the secure computation literature), i.e., both platforms will strictly follow the protocol execution. Second, the number of items whose size/scale is usually large and publicly-known in practice. Therefore, both platforms can agree on an iteration number $T$ such that $T < n$,

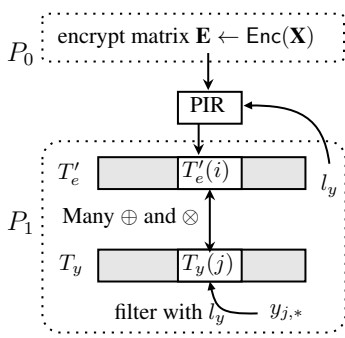

Figure 6: Matrix multiplication with sensitive sparsity.

before running our proposed framework. Each platform can shut down the program if it reaches the pre-defined number of iterations. Moreover, the reveal of the social term to $P_0$ could be avoided by taking the whole model training procedure as an MPC functionality and designing a complicated protocol for it. Inevitably, such protocol introduces impractical communication costs, and we leave how to solve this efficiently as a future work.

## 4  Experiments

Our experiments intend to answer the following questions. **Q1:** How do the social recommendation models using both rating data on $P_0$ and social data on $P_1$ outperform the model that only uses rating data on $P_0$ (Section 4)? **Q2:** How does our model perform compared with SeSoRec (Section 4)? **Q3:** How does the social data sparsity affect the performance of SeSoRec and our model (Section 4)?

**Implementation and setup.** We run our experiments on a machine with 4-Core 2.4GHz Intel Core i5 with 16G memory, we compile our program using a modern C++ compiler (with support for C++ standard 17). In addition, our tests were run in a local network, with $\approx 3$ms network latency. For additive HE scheme, we choose the implementation of libpaillier[1]. Also, we use Seal-PIR[2] with same parameter setting as the original paper [1]. For security, we choose 128-bit computational security and 40-bit statistical security as recommended by NIST [2]. Similarly we leverage the generic ABY library[3] to implement SeSoRec [5] and MPC building blocks such as addition, multiplication, and truncation. In particular, we choose 64-bit secret sharing in all our experiments.

---

[1]libpaillier: http://acsc.cs.utexas.edu/libpaillier/, GPL license

[2]Seal-PIR: https://github.com/microsoft/SealPIR, MIT license

[3]ABY: https://github.com/encryptogroup/ABY, LGPL license

**Dataset.** We choose two popular benchmark datasets to evaluate the performance of our proposed model, i.e., Epinions [19] and LibraryThing (Lthing) [32], both of which are popularly used for evaluating social recommendation tasks. Following existing work [5], we remove the users and items that have less than 15 interactions for both datasets. We summarize the statistics of both datasets after process in Table 1. Notice that we assume users' rating data are located at $P_0$, users' social data are located at $P_1$, and $P_0$ and $P_1$ share the same user set.

Table 1: Dataset statistics.

| Dataset | #user | #item | #rating | rating density | #social relation | social density |
|---------|-------|-------|---------|----------------|------------------|----------------|
| Epinions | 11,500 | 7,596 | 283,319 | 0.32% | 275,117 | 0.21% |
| Lthing | 15,039 | 14,957 | 529,992 | 0.24% | 44,710 | 0.02% |

**Comparison Methods.** We compare $S^3$Rec with the following classic and state-of-the-art models:

– *MF* [20] is a classic matrix factorization model that only uses rating data on $P_0$, i.e., when $\gamma = 0$ for $S^3$Rec.
– *Soreg* [18] is a classic social recommendation model, which does not consider data privacy and assumes both rating data and social data are available on $P_0$.
– *SeSoRec* [5] tries to solve the privacy-preserving cross-platform social recommendation problem, but suffers from security and efficiency problem.

**Hyper-parameters.** For all the model, during comparison, we set $k = 10$. We tune learning rate $\theta$ and regularizer parameter $\lambda$ in $\{10^{-3}, 10^{-2}, ..., 10^1\}$ to achieve their best values. We also report the effect of $K$ on model performance.

**Metrics.** We will evaluate both accuracy and efficiency of our proposed model. For accuracy, we choose Root Mean Square Error (RMSE) as the evaluation metric, since ratings range in [0, 5]. For efficiency, we report the computation time (in seconds) and the communication size between $P_0$ and $P_1$ (in gigabytes), if has, for all the models. We use five-fold cross-validation during experiments.

**Performance Comparison.** We first compare the model performances in terms of accuracy (RMSE) and efficiency (total time and communication). Table 2 shows the time and communication for each epoch, where time is shown in seconds, and communication is shown in GB.

From those Tables, we find that: (1) the use of social information can indeed improve the recommendation performance of the rating platform, e.g., 1.193 vs. 1.062 and 0.927 vs. 0.098 in terms of RMSE on Epinions and Lthing, respectively. This result is consistent with existing work from [18, 5]; (2) despite the same RMSE as SeSoRec and Soreg, $S^3$Rec significantly improves the efficiency of SeSoRec, especially on the more sparse Lthing dataset, reducing the total time for one epoch from around 4.5 hours to around 4.5 minutes, and reducing the total communication from nearly 1.3TB to around 2.2GB. This yields an improvement of $18.57\times$ faster, and $224.8\times$ less communication on Epinions and $61.37\times$ faster and $620.2\times$ less communication on Lthing, respectively.

**Effect of Social Data Sparsity.** Next, we try to study the effect of social data sparsity on training efficiency. In order to do this, we sample the social relation of both datasets with a rate of 0.8, 0.6, and 0.4. As the result, the RMSEs of both SeSoRec and $S^3$Rec decrease to 1.0932, 1.1373, 1.1751 on Epinions dataset, and 0.9112, 0.9187, 0.9210 on Lthing dataset. The rational behind is that recommendation performance decreases with the number of social relations. We also report the efficiency of both models on Epinions and Lthing datasets in Table 3. From it, we can find that the computation time and communication size of SeSoRec are constant no mater what the sample rate is. In contrast, the computation time and communication size of $S^3$Rec decrease linearly with sample rate. This result benefits from that $S^3$Rec can deal with sparse social data with our proposed sparse matrix multiplication protocols.

**Effect of $k$.** For efficiency, we report the running time and communication size of SeSoRec and PriorRec w.r.t $k$ in Table 4, where we use the Epinions dataset. From it, we can get that in average, $S^3$Rec improves SeSoRec **18.6x** in terms of total running time and **225x** in terms of communication. More specifically, we observe that (1) the total running time of both SeSoRec and PriorRec increase

Table 2: Comparison results of different models in terms of model accuracy (in RMSE), running time (in seconds), and communication size (in GB), on Epinions and Lthing datasets.

| Models | Epinions dataset | | | | Lthing dataset | | | |
|---|---|---|---|---|---|---|---|---|
| | MF | Soreg | SeSoRec | S$^3$Rec | MF | Soreg | SeSoRec | S$^3$Rec |
| RMSE | 1.193 | 1.062 | 1.062 | 1.062 | 0.927 | 0.908 | 0.908 | 0.908 |
| Offline Time | - | - | 7,271 | **10.86** | - | - | 14,450 | **8.912** |
| Total Time | 3.846 | 40.50 | 7,799 | **419.9** | 9.596 | 57.76 | 16,084 | **262.1** |
| Offline Comm. | - | - | 788.3 | **0** | - | - | 1,348 | **0** |
| Total Comm. | - | - | 798.6 | **3.552** | - | - | 1,365 | **2.201** |

Table 3: Comparison results by varying social data sparsity on Epinions and Lthing datasets.

| Metric | Models | Epinions | | | Lthing | | |
|---|---|---|---|---|---|---|---|
| | | 0.4 | 0.6 | 0.8 | 0.4 | 0.6 | 0.8 |
| Total time (Seconds) | SesoRec | 7,799 | 7,799 | 7,799 | 16,084 | 16,084 | 16,084 |
| | S$^3$Rec | 366.3 | 381.2 | 401.8 | 194 | 217 | 238 |
| | (Improvement) | (21.29x) | (20.46x) | (19.41x) | (82.91x) | (74.12x) | (67.58x) |
| Total communication (GB) | SesoRec | 798 | 798 | 798 | 1,366 | 1,366 | 1,366 |
| | S$^3$Rec | 3.12 | 3.29 | 3.46 | 1.62 | 1.82 | 2.01 |
| | (Improvement) | (255x) | (243x) | (231x) | (843x) | (751x) | (680x) |

Table 4: Effect of $k$ on running time and communication size on Epinions dataset.

| Models | SeSoRec | | | S$^3$Rec | | |
|---|---|---|---|---|---|---|
| | $k = 10$ | $k = 15$ | $k = 20$ | $k = 10$ | $k = 15$ | $k = 20$ |
| Offline Time | 7,271 | 12,651 | 17,676 | 10.86 | 9.667 | 9.815 |
| Total Time | 7,799 | 13,565 | 19,585 | 419.9 | 449.6 | 527.4 |
| Offline Comm. | 788.3 | 1,182 | 1,577 | 0 | 0 | 0. |
| Total Comm. | 798.6 | 1,198 | 1,597 | 3.552 | 3.552 | 3.552 |

with $k$, but the increase rate of S$^3$Rec is slower than that of SeSoRec; (2) the communication size of SeSoRec increases with $k$, in contrast, the communication size of S$^3$Rec is constant. This result demonstrates that our proposed S$^3$Rec has better scalability than SeSoRec in terms of both running time and communication size.

## 5  Related Work

Traditional recommender systems that only consider user-item rating information suffer from severe data sparsity problem [20]. On the one hand, researchers extensively incorporate other kinds of information, e.g., social [27], review [25], location [16], and time [7], to further improve recommendation performance. On the other hand, existing studies begin to explore information on multiple platforms or domains to address the data sparsity problem in recommender systems, i.e., cross-platform and cross-domain recommendation [17, 34, 33]. However, most of them cannot solve the data isolation problem in practice.

So far, there has been several work that may be applied for privacy-preserving cross-domain recommendations. For example, [23] applied garbled circuits for secure matrix factorization, and it has high security but low efficiency. Chai et al. [4] adopted homomorphic encryption for federated matrix factorization, but it assumes the existence of a semi-honest server and is not provable secure. [13] uses differential privacy to protect user location privacy using transfer learning technique, which is not provable secure and does not suitable to our problem. The most similar work to ours is SeSoRec [5],

however, it suffers from two main shortcomings: (1) as admitted by SeSoRec, it improves efficiency by sacrificing security. That is, it reveals the sum of two rows or two columns of the input matrix. We emphasis that this raises serious security concern in the social recommendation since one may infer detailed social relations from the element-wise sum of two rows/columns of the user social matrix, especially when social relations are binary values; (2) SeSoRec treats the social data as a dense matrix and thus still has serious efficiency issue under the practical sparse social data setting.

## 6 Conclusion

This paper aims to solve the data isolation problem in cross-platform social recommendation. To do this, we propose $S^3$Rec, a sparsity-aware secure cross-platform social recommendation framework. $S^3$Rec conducts social recommendation task and preserves data privacy at the same time. We also propose two secure sparse matrix multiplication protocols to improve the model training efficiency. Experiments conducted on two datasets demonstrate that $S^3$Rec improves the computation time and communication size by around $40\times$ and $423\times$ on average, compared with the state-of-the-art work.

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
