# Appendix: Exploiting Data Sparsity in Secure Cross-Platform Social Recommendation

**Jamie Cui**[1], **Chaochao Chen**[2,1*], **Lingjuan Lyu**[3], **Carl Yang**[4], and **Li Wang**[1]

[1]Ant Group
[2]Zhejiang University
[3]Sony AI
[4]Emory University
[*]Corresponding author, email: `zjuccc@zju.edu.cn`

## A  Private Information Retrieval (PIR)

Our protocol requires a bandwidth-efficient single-server PIR scheme. Specifically, in our work, we use the famous Seal PIR [1], which additionally leverages levelled homomorphic encryption scheme Fan-Vercauteren (FV) [2]. Slightly different from the standard PIR scheme, Seal-PIR further allows the client to send a compressed query to the server, which is then decompressed on the server by PIR.Expand algorithm. In general, Seal-PIR consists of four algorithms (see Figure 1):

$$(\mathsf{PIR.Query}, \mathsf{PIR.Expand}, \mathsf{PIR.Response}, \mathsf{PIR.Extract}).$$

In particular, we consider single-server PIR with a computationally bounded adversary. The security definition is given below,

**Definition 1** (Security of Computational PIR). *Let $\mathcal{F}_{\mathsf{PIR}}$ be the query function which takes input as an index in $[|\mathsf{DB}|]$, where $\mathsf{DB}$ is the target database, and let $r$ be the user's random coins. Then for every distinct indices $i, j \in [|\mathsf{DB}|]$, and for every probabilistic polynomial-time adversary $\mathcal{A}$ bounded by security parameter $\lambda$,*

$$|\Pr_r[\mathcal{A}(1^\lambda, \mathcal{F}_{\mathsf{PIR}}(i, r)) = 1] - \Pr_r[\mathcal{A}(1^\lambda, \mathcal{F}_{\mathsf{PIR}}(j, r)) = 1]| \le \mathsf{negl}(\lambda).$$

Since the construction of Seal-PIR replies on FV, now we show the details of the FV scheme.

**Fan-Vercauteren (FV).** FV is a levelled homomorphic encryption scheme, where plaintexts are represented as polynomials of degree at most $N$, and integer coefficients modulo $t$. More specifically, the plaintext polynomials are from the quotient ring $R_t = \mathbb{Z}_t[x]/(x^N + 1)$, where $N$ is a power of 2, and $t$ is the plaintext modulus that determines how many data can a single FV plaintext represents. The ciphertexts in FV comprises of two polynomials from ring $R_q = \mathbb{Z}_q[x]/(x^N + 1)$, where $q$ is the coefficient modulus affecting the noise budget of a ciphertext and the security level of the entire cryptosystem. As operations such as addition or multiplication are performed, the noise of the output ciphertext grows based on the noise of the operands and the operation being performed. For the purpose of PIR, we care about the following operations:

$$p(x^k) = \mathsf{Sub}_k(c),$$

where $c$ is a ciphertext encrypting a polynomial $p(x)$, and $k$ is an odd integer. For instance, if $c$ encrypts the polynomial $p(x) = x^3 + 4x$, $\mathsf{Sub}_2(c)$ returns $(x^2)^3 + 4x^2 = x^6 + 4x^2$.

## B  Security Proofs

First, we give the formal definitions of computational indistinguishability, simulation-based security (which is a popular tool for proving secure computation security), and IND-CPA security (indis-

---

Parameters: $d \in [1, \log n], m = n^{1/d}$, compression $c \in [0, \log_2 N]$.

PIR.Query$(i)$

1 : Generate $s_j = (s_{j,i})_{i \in [m]}$, the $d$-th selections vector in $\{0,1\}^m$

2 : $l \leftarrow \lceil d \cdot m / 2^c \rceil$

3 : Parse $(s_1, ..., s_d)$ as $(s'_1, ..., s'_d)$ vectors in $\{0,1\}^{2^c}$

4 : $\forall j \in [l], m_j \leftarrow \sum_{i \in 2^c} (2^{-c} \mod t) \cdot s_j[i] \cdot x^i \in R_t$

5 : $\forall j \in [l], q_j \leftarrow \mathsf{Enc}_{sk}(m_j)$

6 : $\vec{q} = (q_j)_{j \in [l]} \in R_q^l$

7 : **return** $\vec{q}$

PIR.Expand$(q)$

1 : $l \leftarrow \lceil d \cdot m / 2^c \rceil$

2 : ciphertexts $\leftarrow []$

3 : **for** $j \in \{1, ..., l\}, \mathsf{ctxts} = [q_j = (c_0, c_1)]$ **do**

4 :    **for** $a \in \{0, ..., c-1\}$ **do**

5 :       **for** $b \in \{0, ..., 2^a - 1\}$ **do**

6 :          Let $c'_k \leftarrow c_0 + \mathsf{Sub}_{2^{c-a}+1}(c_0)$

7 :          Let $c'_{k+2^a} \leftarrow c_1 + \mathsf{Sub}_{2^{c-a}+1}(c_1)$

8 :       **endfor**

9 :       $\mathsf{ctxts} = [c'_0, ..., c'_{2^{a+1}-1}]$

10 :    **endfor**

11 :    ciphertexts $\leftarrow$ ciphertexts$\|$ctxts

12 : **endfor**

13 : Let $\vec{s} \leftarrow$ ciphertexts

14 : **return** $\vec{s}$

PIR.Response$(\mathsf{DB}, \vec{s})$

1 : $\forall s_i \in \vec{s}$, let $\mathsf{DB}_i = \bigoplus_{j=0}^{m} \mathsf{DB.Get}(i,j) \otimes s_i[j]$

2 : **return** $r := \mathsf{DB}_i$

PIR.Extract$(r)$

1 : $\forall i \in \{1, ..., d\}$, let $r := \mathsf{Dec}_{sk}(r) \mod 2^l$

2 : **return** $\mathsf{DB}_i := r$

---

Figure 1: Seal-PIR algorithms, where we denote the number of entries in the database as $n$, and a user-defined recursion level as $d$. Also we use $\otimes$ to denote the homomorphic operation of plaintext-ciphertext multiplication $\oplus$ to denote the homomorphic operation of ciphertext addition. We also use $\mathsf{DB.Get}(i,j)$ to indicate get the $i$th dimension vector's $j$th value.

tinguishable chosen-plaintext attack). Then we use these definitions to prove lemma 1 and lemma 2.

## B.1 Definitions and Tools

**Definition 2** (Computational Indistinguishability). *Let $a \in \{0,1\}^*$ be the inputs from participants, and $\lambda \in \mathbb{N}$ be the security parameter, two probability functions $\{\mathcal{F}_0(a, \lambda)\}_{a \in \{0,1\}^*, \lambda \in \mathbb{N}}$ and*

$\{\mathcal{F}_1(a, \lambda)\}_{a \in \{0,1\}^*, \lambda \in \mathbb{N}}\}$ *are said to be computational indistinguishable, if for every non-uniform polynomial-time algorithm $\mathcal{A}$, there exits a negligible function* $\mathsf{negl}(\lambda)$, *such that for every* $a \in \{0,1\}^*$ *and every* $\lambda \in \mathbb{N}$,

$$|\Pr[\mathcal{A}(\mathcal{F}_0(a, \lambda)) = 1]| - |\Pr[\mathcal{A}(\mathcal{F}_1(a, \lambda)) = 1]| \leq \mathsf{negl}(\lambda).$$

**Definition 3** (Simulation-based Security). *Let* $\mathcal{F} = (\mathcal{F}_0, \mathcal{F}_1)$ *be a functionality, we say a protocol* $\pi$ *securely computes* $\mathcal{F}$ *in the presence of static semi-honest adversaries if there exists a probabilistic polynomial-time algorithm* $\mathcal{S}_0$ *and* $\mathcal{S}_1$ *such that*

$$\{(\mathcal{S}_0(1^\lambda, x, \mathcal{F}_0(x, y)), \mathcal{F}(x, y))\} \overset{c}{\equiv} \{(\mathsf{view}_0^\pi(\lambda, x, y), \mathsf{output}^\pi(\lambda, x, y))\},$$
$$\{(\mathcal{S}_1(1^\lambda, y, \mathcal{F}_1(x, y)), \mathcal{F}(x, y))\} \overset{c}{\equiv} \{(\mathsf{view}_1^\pi(\lambda, x, y), \mathsf{output}^\pi(\lambda, x, y))\},$$

*where* $x, y$ *are inputs from* $P_0$ *and* $P_1$ *separately.*

**Definition 4** (IND-CPA Security Game). *A public-key encryption scheme is said to be IND-CPA secure if for all probabilistic polynomial-time adversary* $\mathcal{A}$,

$$\Pr[\mathsf{Game}^{\mathcal{A}}(1^\lambda) = 1] \leq \frac{1}{2} + \mathsf{negl}(\lambda),$$

*where the IND-CPA*$^{\mathcal{A}}(\lambda)$ *is defined as follows:*

| IND-CPA$^{\mathcal{A}}(\lambda)$ |
| --- |
| 1 : $\quad b \leftarrow \$ \{0, 1\}$ |
| 2 : $\quad (\mathsf{pk}, \mathsf{sk}) \leftarrow \$ \mathsf{KGen}(1^\lambda)$ |
| 3 : $\quad (\mathsf{state}, m_0, m_1) \leftarrow \$ \mathcal{A}(1^\lambda, \mathsf{pk}, c)$ |
| 4 : $\quad c \leftarrow \$ \mathsf{Enc}(\mathsf{pk}, m_b)$ |
| 5 : $\quad b' \leftarrow \$ \mathcal{A}(1^\lambda, \mathsf{pk}, c\mathsf{state})$ |
| 6 : $\quad \mathbf{return}\ b' = b$ |

## B.2 Proofs of Lemma 1 and Lemma 2

**Lemma 1.** *The protocol in Figure 5 (i.e. Dense-Sparse Matrix Multiplication with Insensitive Sparsity) is secure in the MPC-hybrid model.*

*Proof.* We use simulation-based proof [3] for Lemma 1. Since we are considering our protocol in the MPC-hybrid model, we assume there are two ideal MPC functionalities $\mathcal{F}_{\mathsf{mul}}$ and $\mathcal{F}_{\mathsf{add}}$, where $\mathcal{F}_{\mathsf{mul}}$ takes $[x]_0, [y]_0$ from $P_0$, and $[x]_1, [y]_1$ from $P_1$, and returns $[z]_0$ to $P_0$ and $[z]_1$ to $P_1$ such that $z = xy$,. Similarly, $\mathcal{F}_{\mathsf{add}}$ takes $[x]_0, [y]_0$ from $P_0$, and $[x]_1, [y]_1$ from $P_1$, and returns $[z]_0$ to $P_0$ and $[z]_1$ to $P_1$ such that $z = x + y$. The security of $\mathcal{F}_{\mathsf{mul}}$ and $\mathcal{F}_{\mathsf{add}}$ says that $[z]_0$ and $[z]_1$ are indistinguishable from uniform randomness from the sole view of $P_0$ or $P_1$ separately. That is to say, for all $c \in \mathbb{Z}_\delta$

$$|\Pr[[\mathcal{F}_{\mathsf{mul}}([x], [y])]_i = c] - \Pr[r = c]| = 0,$$
$$|\Pr[[\mathcal{F}_{\mathsf{add}}([x], [y])]_i = c] - \Pr[r = c]| = 0,$$

where $i \in \{0, 1\}$, $\delta$ is the share bit length, and $r$ is a uniform random number sampled from $\mathbb{Z}_\delta$.

Therefore, we need to build a simulator $\mathcal{S}_0$ to simulate the view of $P_0$ in protocol $\pi$. Note in our protocol, all the interactions between $P_0$ and $P_1$ come from $\mathcal{F}_{\mathsf{mul}}$ and $\mathcal{F}_{\mathsf{add}}$, then we can use $\mathcal{S}_0$ to generate a uniform random value $r \in \mathbb{Z}_\delta$ for each stage independently. Given that the outputs of ideal functionalities $\mathcal{F}_{\mathsf{mul}}$ and $\mathcal{F}_{\mathsf{add}}$ are indistinguishable from uniform randomness for both view of the parties, this simulation works for both $\mathcal{S}_0$ and $\mathcal{S}_1$, and the simulation completes. $\qquad\square$

**Lemma 2.** *The protocol in Figure 6 (i.e. Dense-Sparse Matrix Multiplication with Sensitive Sparsity) is secure in the PIR-hybrid model with the leakage of* $|l_y|$.

*Proof.* We also use simulation-based proof [3] for Lemma 2. First, we assume there is a ideal functionality $\mathcal{F}_{\mathsf{pir}}$, which takes a value set DB from $P_0$, an index $i$ from $P_1$, and returns $\mathsf{DB}_i$ to $P_1$. The security of $\mathcal{F}_{\mathsf{pir}}$ are biased since PIR only aim to hide the query $i$ from $P_0$. More formally, the security of single-database computational PIR says that, for every distinct $i, j \in [n]$ and every probabilistic polynomial-time (PPT) adversary $\mathcal{A}$,

$$|\Pr[\mathcal{A}(1^\lambda, \mathcal{F}_{\mathsf{pir}}(i, r)) = 1] - \Pr[\mathcal{A}(1^\lambda, \mathcal{F}_{\mathsf{pir}}(j, r)) = 1]| \leq \mathsf{negl}(\lambda),$$

where $\lambda$ is the computational security parameter, $n$ is the size of the database DB, and $\mathsf{negl}$ is a negligible function bounded by computational security parameter $\lambda$.

Since in the offline phase, $P_0$ only sends the public key $pk$ to $P_1$, the simulation for offline phase is trivial. For online phase, we build simulator $\mathcal{S}_0$ and $\mathcal{S}_1$ for $P_0$ and $P_1$ separately.

**Simulator** $\mathcal{S}_0$: First, for step 1-3, $P_0$ and $P_1$ invokes $\mathcal{F}_{\mathsf{pir}}$ $|l_y|$ times (which is the pre-defined leakage). $\mathcal{S}_0$ can simulate this process by inputing $\mathcal{F}_{\mathsf{pir}}$ with $|l_y|$ random encryptions, i.e. $\mathsf{DB} = \{\mathsf{Enc}_{pk}(r_1), ..., \mathsf{Enc}_{pk}(r_n)\}$, where $r_1, .., r_n \in \mathbb{Z}_\delta$ are uniform random numbers. For the view of $\mathcal{S}_0$, IND-CPA security says that with a PPT adversary $\mathcal{A}$ who can access the encryption oracle,

$$|\Pr[\mathcal{A}(1^\lambda, \mathsf{Enc}_{pk}(\mathsf{DB}_i)) = 1] - \Pr[\mathcal{A}(1^\lambda, \mathsf{Enc}_{pk}(r)) = 1]| \leq \mathsf{negl}(\lambda).$$

Therefore, from the view of $\mathcal{S}_0$ is indistinguishable from a random encryption with leakage of $|l_y|$. For the rest of the protocol, party $P_0$ sends nothing. The simulation completes.

**Simulator** $\mathcal{S}_1$: First, for step 1-3, $\mathcal{S}_1$ invokes $\mathcal{F}_{\mathsf{pir}}$ $|l_y|$ times. Assuming the secure instantiation of $\mathcal{F}_{\mathsf{pir}}$, the simulation for $\mathcal{S}_1$ in step 1-3 is trivial. Then at step 4, $\mathcal{S}_1$ generates a random value $r \in \mathbb{Z}_\delta$, encrypts it with $P_0$'s public key $\mathsf{Enc}_{pk}(r)$, and sends the result to $P_0$. Observe that for all $c \in \mathbb{Z}_\delta$, for a given $m \in \mathbb{Z}_\delta$,

$$|\Pr[r = c] - \Pr[m - r = c]| = 0.$$

The simulation completes. $\qquad\qquad\qquad\qquad\qquad\qquad\qquad\qquad\qquad\qquad\qquad\qquad\qquad\square$