# OpenReview forum: "Exploiting Data Sparsity in Secure Cross-Platform Social Recommendation"
_NeurIPS.cc/2021/Conference — NeurIPS 2021 Poster_

### Official Review · Reviewer_Maoq · 2021-07-11

**Rating:** 4
**Confidence:** 2

**Summary:**

This paper proposes a secure and efficient cross-platform social recommendation algorithm. It solves a standard matrix factorization problem using network information, in which one platform has the rating data and another has the social information.


**Ethical Concerns:**

The authors have adequately addressed the limitations and there is no potential negative societal impact of their work.

**Limitations And Societal Impact:**

None.

**Main Review:**

Strengths:

The paper is well-written ad easy to follow.

Comments:

(1) My first comment on the paper is about the problem setup. I think the paper needs justifications to (1) demonstrate the practicality (how reasonable it is). Specifically, I wonder how reasonable it is to assume that two stakeholders store social and rating information separately. Realistic examples will be helpful to demonstrate the actual value of the problem formulation. I think it is more reasonable to assume that two platforms have the same information on items, e.g., Spotify, youtube, and pandora have information on similar music.

(2)  Still related to the problem setup, I think there are security issues related to the problem setup. Why and how can two platforms match this sensitive information on user identity? To do so requires the platform to know some identifiable information and that they will be able to communicate through this unique identifier.

(3) There are other matrix factorization methods, such as svd++ and different deep learning-based approaches. I suspect that even without social information, these methods may be able to achieve better performance. If this is the case, it is not clear this stylized extension on matrix factorization to obtain security is useful or not. I think the experiment section can be enriched by comparing with more benchmarks (without network information).

(4) Regarding the communication mechanism in section 3.1, it is unclear how it can communicate the gradients in the second part of eq (2) without security concern. Specifically, for each iteration, there are no gradients and n(n-1)/2 entries. Suppose there are T iterations in total, when k * T ≥ (n-1)/2, it is possible for P0 to recover social influence in P1. There may be things I miss; please correct me if I am wrong.

**Time Spent Reviewing:**

3.5

---

> ### Author Response · Authors · 2021-08-10
> **Response to Reviewer Maoq**
>
> Thanks for your thoughtful comments.  We hope to address your concerns through the following response.
>
> 1. This data isolation setup is quite common in practice. As we have introduced in our paper (line 20), say Netflix and Facebook want to jointly train a recommendation model, those training data (i.e. user-item rating data and user-user social data) are naturally generated or collected by Netflix and Facebook, separately. The intuition behind this setup is that Netflix usually has less social data than Facebook, and those social data in Facebook can improve recommendation performance. More generally, the problem setup is suitable for an e-commerce platform and a social platform that want to jointly train a better recommendation model in practice.
>
> 2. Yes, there are security issues related to the problem setup, but fortunately there are existing cryptographic tools (e.g. Private Set Intersection protocols [1]) to protect sensitive identity information. Again, say that Facebook and Netflix want to collaboratively train a social recommendation model, how do they figure out the matched users? Intuitively, they can priorly agree on one unique identifier which they both have (e.g. phone number or email), and now the question is how to find the common identifier between two sets $X_f$ and $X_n$. As a result, both parties only learn the matched users but nothing else. This problem is known as the private set intersection problem in literature, and many efficient and scalable protocols have been proposed to do so, which is out of the scope of our paper. However, we are happy to add these descriptions in our final version.
>
> 3. Indeed, it might be the case that a certain complicated model would outperform our proposed framework. However, this is data dependent and case dependent. It is common sense that additionally considering social information is effective in improving recommendation performance. Besides, the main contribution of our paper is to show the feasibility of exploiting data sparsity on a specific secure recommendation model to improve efficiency. Through our work, we want to shed light on a possible future direction of this research area. To the best of our knowledge, there is no secure NN-based recommendation model yet, therefore we leave the demonstration of complicated models (e.g., deep learning-based approaches) in future work.
>
> 4. As we claimed in Section 3.2, in real cases, the number of iterations (i.e., $T$) is much smaller than the number of variables (i.e., $n$). For example, in our experiments, $T=100$ and $n>10^4$.  Meanwhile, in practice, the social platform can always control the iteration rounds “$T$” to ensure the security of its social data (by sending less than $T$ social terms to the rating platform). As we have pointed out in the security discussions, we leave better solutions for future work.
>
> [1] Kolesnikov V, Kumaresan R, Rosulek M, et al. Efficient batched oblivious PRF with applications to private set intersection. CCS, 2016.

---

> > ### Comment · Reviewer_Maoq · 2021-08-20
> > **Further comments.**
> >
> > Thank you for the explanations for 1 & 2; these responses help with motivating the problem setup. It will be helpful to add these points to the revised paper.
> >
> > 3. I agree that it has been widely-observed that adding social network help with improving the recommendation quality. However, the empirical comparisons in the literature are based on the same model. If the model itself is different, it is not clear whether adding information to a shallow model would outperform a more complicated deep learning model. I think this is an important issue to address. Take the Netflix example: if Netflix can achieve superior performance with a complicated NN model, it would not be interested in partnering with FB to build a simple model and integrate network information. Since this question has also been brought up by d4vk and FftY, I wonder if it's possible to address the two following aspects.
> >
> >  (a) SVD++ is shown to perform very competitively to deep learning models. Could you discuss how your model can generalize to SVD++?
> >  (b) As you have responded to d4vk, NN training is a hot topic. Could you please discuss more specifically how other researchers working specifically on this area can learn from your paper? What are the theoretical and practical implications?
> >
> > 4. For the platform with the social network, it is easy to satisfy T < n. I wonder if T < n is still easy to satisfy?

---

> > > ### Author Response · Authors · 2021-08-24
> > > **Response to Reviewer Maoq**
> > >
> > > **How can our model generalize to SVD++?**
> > >
> > > Thanks for asking the suggestive question. We simply summarize how our proposed framework generalizes to SVD++ as follows.
> > >
> > > First of all, as we have shown in line 133 that 'Factorization based social recommendation models are the combination of a “basic factorization model” and a “social information model”.’ As a specific implementation in our paper, we choose matrix factorization (MF) as the basic factorization model, and a social regularization term as the social information model.
> > >
> > > Second, simply speaking, one can replace MF with SVD++ as the basic factorization model, to generalize our proposed framework to SVD++. Specifically, the objective function in our paper is
> > > $$
> > > \mathcal{L}= \frac{1}{2}I_{i,j}\left(r_{i,j} - \textbf{u}\_{\ast, i}^T \textbf{v}\_{\ast, j} \right)^2 + \frac{\lambda}{2} \sum\limits_{i=1}^{m} \|\textbf{u}\_{\ast, i}\|\_F^2 + \frac{\lambda}{2} \sum\limits_{j=1}^{n} \|\textbf{v}\_{\ast, j}\|\_F^2 + \frac{\gamma}{2}\sum\limits_{i=1}^{m} \sum\limits_{f=1}^{m} s_{i,f} \|\textbf{u}\_{\ast, i} - \textbf{u}\_{\ast, f} \|\_F^2,
> > > $$
> > > where the predicted rating is defined as $\hat{r}\_{i,j}=\textbf{u}\_{\*,i}^T\textbf{v}\_{\*,j}$. In contrast, the predicted rating in SVD++ is defined as
> > > $$
> > > \hat{r}\_{i,j}=\mu +b_i + b_j+v_j^T\left(u_i+|N(i)|^{-\frac{1}{2}}\sum_{k\in N(i)} y_k\right)
> > > $$
> > > where $\mu$ is the overall average rating, $b_i$ and $b_j$ indicate the observed deviation of user $i$ and item $j$, respectively, from the average (the same as [1]). Thus, when applying our solution in SVD++, the loss function becomes
> > > $$
> > > \mathcal{L}=\sum\limits_{i=1}^{m} \sum\limits_{j=1}^{n}  \frac{1}{2}I_{i,j}\left(r_{i,j} - \hat{r}\_{i,j} \right)^2+ \frac{\lambda}{2} b_{i}^2+ \frac{\lambda}{2} b_{j}^2+ \frac{\lambda}{2} \sum\limits_{i=1}^{m} \|\textbf{u}\_{\*, i}\|\_F^2+ \frac{\lambda}{2} \sum\limits_{i=1}^{m} \|\textbf{y}\_{\*, i}\|\_F^2+ \frac{\lambda}{2} \sum\limits_{j=1}^{n} \|\textbf{v}\_{\*, j}\|\_F^2+ \frac{\gamma}{2}\sum\limits_{i=1}^{m} \sum\limits_{f=1}^{m} s_{i,f} \|\textbf{u}\_{\*, i} - \textbf{u}\_{\*, f} \|\_F^2,
> > > $$
> > > where $N(i)$ denotes all items for which user $i$ provided an implicit preference, and $|N(i)|^{-\frac{1}{2}}\sum_{k\in N(i)} y_k$ presents the perspective of implicit feedback. Let $e_{i.j}=r_{ij}-\hat{r}_{i,j}$, we can obtain the following parameter update functions,
> > > - $\partial\mathcal{L}/\partial b_i= - I_{i,j}\cdot v_j \cdot e_{i,j} + \lambda b_i$ (calculated by the rating platform)
> > > - $\partial\mathcal{L}/\partial b_j=- I_{i,j}\cdot v_j \cdot e_{i,j} + \lambda b_j$  (calculated by the rating platform)
> > > - $\partial\mathcal{L}/\partial u_i=- I_{i,j}\cdot v_j \cdot e_{i,j} +\lambda  u_i +\text{the social term}$  (calculated by the rating platform and the social platform collaboratively)
> > > - $\partial\mathcal{L}/\partial v_j=- I_{i,j}\cdot v_j \cdot e_{i,j} + \lambda v_j$ (calculated by the rating platform)
> > > - $\partial\mathcal{L}/\partial y_k=- I_{i,j}\cdot v_j \cdot e_{i,j}  \cdot |N(i)|^{-\frac{1}{2}} + \lambda y_k$  (calculated by the rating platform)
> > >
> > > From the above derivation, we can conclude that the key is securely calculating $\partial\mathcal{L}/\partial u_i$, which can be done similarly as our solution to the social term.
> > >
> > > **How can state-of-the-art (STOA) secure NN frameworks learn from our work? What are the theoretical and practical implications?**
> > >
> > > Most STOA secure NN frameworks (including CryptGPU [2], SecureNN [3], Falcon [4], et. al.) focus on improving the protocol efficiency for dense data, few of them can solves the data sparsity problem in secure NN training. Meanwhile, data sparsity popularly exists in NN models and thus NN models naturally contain many sparse matrix multiplications, e.g., in the first layer when input data is sparse and in the hidden layer when its pre-layer performs drop-out. Theoretically, in this paper, we propose two secure sparse matrix multiplication protocols and prove their security and efficiency in our framework. In practice, our proposed secure sparse matrix multiplication protocols (with both insensitive sparsity and sensitive sparsity) can be directly applied in these SOTA secure NN models. Moreover, secure sparse matrix multiplication is quite common in other secure machine learning algorithms besides NN, such as logistic regression, k-means, etc.. Thus our proposed secure sparse matrix multiplication protocols could also be applied to those algorithms to further improve their protocol efficiency.
> > >
> > > **Question about T<n?**
> > >
> > > $T<n$ can be easily satisfied for both the social platform and the rating platform. The reasons are two-folds. First, our proposed framework is secure against a semi-honest adversary (which is a popular threat model in the secure computation literature), i.e., both platforms will strictly follow the protocol execution. Second, $n$ is the number of items whose size/scale is usually large and publicly-known in practice. Thus, both platforms can agree on an iteration number $T$ such that $T<n$, before running our proposed framework. Each platform can shut down the program if it reaches the pre-defined number of iterations.
> > >
> > > **Reference**
> > >
> > > [1] Koren, Yehuda. "Factorization meets the neighborhood: a multifaceted collaborative filtering model." *Proceedings of the 14th ACM SIGKDD international conference on Knowledge discovery and data mining*. 2008.
> > >
> > > [2] Tan, Sijun, et al. "CRYPTGPU: Fast Privacy-Preserving Machine Learning on the GPU." *arXiv preprint arXiv:2104.10949* (2021).
> > >
> > > [3] Wagh, Sameer, Divya Gupta, and Nishanth Chandran. "SecureNN: 3-Party Secure Computation for Neural Network Training." *Proc. Priv. Enhancing Technol.* 2019.3 (2019): 26-49.
> > >
> > > [4] Wagh, Sameer, et al. "Falcon: Honest-majority maliciously secure framework for private deep learning." *arXiv preprint arXiv:2004.02229* (2020).

---

> > > > ### Comment · Reviewer_Maoq · 2021-09-01
> > > > **Thank you for the responses.**
> > > >
> > > > Thank you for adding new results/derivations on SVD++. The response about T < n is also convincing.
> > > >
> > > > It will be good to comment that state-of-art methods do not deal with dense data. I understand that there are no methods that directly deal with sparse matrix completion, hence the paper cannot add a comparison. In the revised manuscript, it will be helpful to add more details on how the method can be directly applied to SOTA secure NN model and other ML algorithms.
> > > >
> > > > In the meantime, I meant to ask the performance comparison between the current method with other state-of-the-art DL methods without using social network information. Still using the Netflix example, if Netflix can achieve good performance using DL methods only with rating information, Netflix will not have an incentive to collaborate with Facebook. However, as in my previous comment, being able to use SVD++ will reduce the performance gap between DL methods and a shallow learning method.
> > > >
> > > > Given the clarifications and new results on SVD++, I'd be happy to adjust my rating to 6.

---

> > > > > ### Author Response · Authors · 2021-09-02
> > > > > **Response to Reviewer Maoq**
> > > > >
> > > > > Thanks for the suggestions. We will add additional content in our revised manuscript on how our proposal can be directly applied to SOTA secure NN model and other ML algorithms.
> > > > >
> > > > > We compare our method with MF, MF with ‘social term’ (i.e., Soreg and S$^3$Rec), SVD++, SVD++ with ‘social term’, and report their RMSE on Epinions dataset in the following table.
> > > > >
> > > > > |         |   MF | MF with social (S$^3$Rec) | SVD++ | SVD++ with social |
> > > > > |---------|----------------------|-------------------|------------------------|---------------------|
> > > > > | | 1.193               | 1.062             | 1.058                 |     0.928       |
> > > > >
> > > > > From the above table, we can find that (1) SVD++ can significantly improve the recommendation performance of MF, (2) SVD++ has comparable performance (even slightly better performance) than Soreg and S$^3$Rec, (3) similar as Soreg, additionally incorporating social information into SVD++ can significantly improve recommendation performance, which is consistent with existing research [1].
> > > > >
> > > > > As we have explained previously, a social recommendation model is the combination of a “basic factorization model” and a “social information model”. Although we take a simple MF model as an example of the basic factorization model, our proposed framework can be applied into other models (e.g., SVD++ we have derived previously). Hopefully, our finding would give a new research direction in recommender systems, especially with the data isolation problem becoming more and more severe. We will also add these descriptions in our paper during revision.
> > > > >
> > > > >
> > > > > [1] Guo, G., J. Zhang and N. Yorke-Smith. “TrustSVD: Collaborative Filtering with Both the Explicit and Implicit Influence of User Trust and of Item Ratings.” AAAI (2015).

---

> > > ### Author Response · Authors · 2021-08-29
> > > **Response to Reviewer Maoq**
> > >
> > > Thanks again for your valuable comments. Please let us know if anything is unclear. We truly appreciate this opportunity to improve our work and shall be grateful for any feedback you could give to us.

---

> > > ### Author Response · Authors · 2021-09-01
> > > **Thanks to Reviewer Maoq**
> > >
> > > We thank you again for the valuable and detailed comments, and for your response to our previous explanations.
> > >
> > > We hope that we have adequately addressed your concern about how our method generalizes to svd++, and how to guarantee the security of the social term. We truly appreciate your valuable feedback that helps us further clarify the problem setups of our work.
> > >
> > > Kindly let us know if there is anything unclear. We appreciate your insightful feedback and comments that help us further improve our work.

---

### Official Review · Reviewer_FftY · 2021-07-16

**Rating:** 7
**Confidence:** 4

**Summary:**

In this paper, the authors study the problem of social recommendation, i.e., rating prediction with a user-item rating matrix and a user-user social relation matrix. Specifically, the authors propose a novel privacy-aware method, i.e., sparsity-aware secure cross-platform social recommendation (S^3Rec).  The main idea of S^3Rec is to compute the social term in the objective function of a classic social recommendation method (see Eq.(1)) by two different parties (or platforms) in a privacy-aware manner via homomorphic encryption and private information retrieval.

**Limitations And Societal Impact:**

Yes.

**Main Review:**

1 Originality:
The proposed method, i.e., sparsity-aware secure cross-platform social recommendation (S^3Rec), is new since no previous work use homomorphic encryption (HE) and private information retrieval (PIR) in such a way for jointly computing the social term in a social recommendation method. Moreover, the difference between the proposed method and the most closely related work, i.e., SeSoRec [5], is also well identified, i.e., security and efficiency.

2 Quality:
The proposed model is technically sound, which is also supported by the empirical studies, i.e., the proposed methods is efficient and accurate.

3 Clarity:
Overall, the paper is well presented and is easy to follow. The motivations on leveraging the social connections for rating prediction in a privacy-aware manner are well justified. Sufficient details of the protocals are also included.

4 Significance:
The authors have conducted extensive empirical studies, including 2 datasets, as well as studies about the effectiveness of method with different levels of sparsites. For rating prediction, the basic MF model is known not the state-of-the-art. For social recommendation, SoRec is also no the state-of-the-art. Moreover, it is not clear whether the proposed framework S^3Rec can be applied to more advanced social recommendation methods.


**Time Spent Reviewing:**

2

---

> ### Author Response · Authors · 2021-08-10
> **Response to Reviewer FftY**
>
> Thanks for your supportive comments. Although MF and SoRec are not state-of-the-art, they are the classical models that could demonstrate our technical contributions in this paper. That is, exploiting data sparsity can significantly improve the efficiency of secure social recommendation models. And yes, our proposed framework could be applied to more complicated models, as long as they could be reduced to matrix multiplication and addition operations. Unfortunately, to our best knowledge, there is no secure NN-based recommendation model yet. We leave the demonstration of complicated models in future work.

---

> ### Author Response · Authors · 2021-09-01
> **Thanks to Reviewer FftY**
>
> We thank you for the detailed comments, and for recognizing the potential of our work in other recommendation models.
>
> We hope our response has adequately addressed your concerns regarding experiments with more complicated models. Please note that, based on the discussions with reviewer Maoq, we have verified that our method works for the svd++ model.
>
> Kindly let us know if anything is unclear. We truly appreciate your valuable feedback and comments that help us further improve our work.

---

### Official Review · Reviewer_2odS · 2021-07-16

**Rating:** 9
**Confidence:** 4

**Summary:**

In this paper, the authors focus on improving the efficiency of a specific problem: secure cross-platform social recommendation. They exploit the sparsity of social data by proposing two novel and secure protocols for matrix multiplication. They also provide theoretical results of both the efficiency and security of their protocols. Also, in experiments, the proposed framework --- S^3^Rec improves the communication and computation efficiency of prior work by approximately 40x and 423x.

**Limitations And Societal Impact:**

See main review

**Main Review:**

Data security and privacy is a hot and important research topic. Build secure recommendation models across domains is of great impact in practice. The idea of exploiting data sparsity in secure cross-platform social recommendation is new. Their proposed secure matrix multiplication protocols are novel to me (in the sense that there seem no existing PIR-based sparse matrix multiplication protocols), and the efficiency improvement brought by those two protocols is quite promising.

This work is technically written with thorough theoretical analysis. In particular, they include detailed security proof. Most of the arguments presented in this work are supported by either solid technical analysis or experiment results.

This paper is well-organized, and it clearly states the main problem in the introduction section, while some technical details are pretty dense and required patient reading, especially for the readers who are not experts in S&P area.

To the best of my knowledge, this paper is the first work exploiting the data sparsity in secure cross-platform social recommendation, and their experiments show remarkable improvements (in terms of efficiency) over the prior framework. Although the adopted social recommendation model is a little out-of-date, I believe it points out the feasibility and practicability of large-scale secure social recommendation.

However, I do have some questions about this paper:

1. Is it possible to use a similar method to improve efficiency on more complicated secure social recommendation models? For example, use secure sparse matrix multiplication protocol on a NN-based recommendation model.
2. Can we replace PIR with other cryptographic primitives in a secure sparse matrix multiplication protocol?

And there are some typos:

1. line 305, there has been several works -> work
2. line 106, we present three popular MPC protocols -> two

Minor concerns:

The authors could state more applications of their secure sparse matrix multiplication protocols, as it shows the generalization of their proposed protocols. Also, I personally recommend the authors try to apply more complex social recommendation models such as the nn-models. Though it's known to be difficult to implement MPC-NN protocol, the authors can discuss (in theory) the impact of exploiting data sparsity in NN-based social recommendation models.

In summary, I think the problem this paper aims to solve is important, the proposed technical solution is novel, and the experimental results are impressive. I strongly support the acceptance of this paper.


**Time Spent Reviewing:**

10

---

> ### Author Response · Authors · 2021-08-10
> **Response to Reviewer 2odS**
>
> Thanks for your valuable advice.
>
> 1. It is possible to improve the efficiency of other social recommendation models, since our proposed method contains a general sparse matrix multiplication protocol. Complicated models can be converted to a secure protocol with a similar trick, as long as they could be reduced to matrix addition and multiplication operations. Though exploiting the data sparsity of NN-based social recommendation models is theoretically feasible, there are fundamental difficulties in implementing a practical secure NN-based recommendation model. For instance, the state-of-art secure NN training protocol (CryptGPU, S&P 2021 [1]) takes approximately 10 days to train AlexNet on Tiny ImageNet (100,000 images) with a decent GPU.
>
> 2. Indeed, one can replace PIR with other cryptographic primitives in a secure sparse matrix multiplication protocol. For example, one can leverage Private Set Intersection (PSI) protocols, but current PSI protocols are less efficient. We choose PIR because PIR protocols guarantee sublinear communication complexity.
>
> Also, thanks a lot for pointing out the typos, we will fix them during revision.
>
> [1] Sijun Tan, Brian Knott, Yuan Tian, David J. Wu. CryptGPU: Fast Privacy-Preserving Machine Learning on the GPU. S&P 2021.

---

> ### Author Response · Authors · 2021-09-01
> **Thanks to Reviewer 2odS**
>
> We sincerely thank you for taking the time to review our paper and for the valuable comments.
>
> Kindly let us know whether we have adequately addressed your comments on the potential usage of our proposed secure sparse matrix multiplication on more complicated recommendation models. We truly appreciate your valuable feedback that helps verify the generality of our findings and improve the clarity of our experiments.

---

### Official Review · Reviewer_d4vk · 2021-07-19

**Rating:** 7
**Confidence:** 2

**Summary:**

This paper studies how to bring secure computation to social recommendation system, by leveraging the distributed (or sparse) data on different platforms while preserving their data privacy. The authors analysis the conventional recommender algorithm's demand on social terms, and propose two secure sparse matrix multiplication methods to conduct the computation. They compare their proposed method with two classical baselines with regards to accuracy, running time and communication size, showing competitive performance and more efficient computation.

**Limitations And Societal Impact:**

Yes

**Main Review:**

Strength:
* The paper proposed an efficient approach to social recommendation.

Weakness:
* Experiment. Since the main claim of the paper is its efficiency, in terms of social recommendation, large scale experiments are expected. However the paper only presents results on small scale datasets. Large scale results on datasets such as Amazon produce review are more convincing.
* Recommendation model. As the proposed computation method seems not tied to a specific recommendation model. More results on modern recommendation models will help the argument, such as "Deep Learning Recommendation Model for Personalization and Recommendation Systems" and "Collaborative filtering and deep learning based recommendation system for cold start items".

**Time Spent Reviewing:**

4

---

> ### Author Response · Authors · 2021-08-10
> **Response to Reviewer d4vk**
>
> We thank you for your insightful suggestions, please find our response below.
>
> 1. We did not conduct experiments on large-scale data due to the following two reasons.
>
> - First, the baseline model (SeSoRec) runs too slow on large-scale data. Take a dataset with around $10^6$ users for example, SeSoRec needs about 16 days to finish one training epoch, which makes it difficult for us to conduct large-scale experiments.
>
> - Second, we believe that our experiments on Epinions and Lthing datasets should be sufficient to demonstrate the improvements over the baseline. Besides, we also include a theoretical analysis of the efficiency improvements.
>
> 2. In our paper, we aim to exploit data sparsity for efficiency improvements for secure social recommendation. To demonstrate the improvements brought by our methods, we choose a classical work (SeSoRec) as our baseline. Our proposal is indeed a general framework and is not tight to a specific model. Besides SocialRec, our method can be implemented on any model that could be reduced to matrix multiplication and addition operations. Moreover, although deep learning based recommendation models achieve state-of-the-art performance in plaintext settings, they are too expensive for secure MPC protocols since they contain many non-linear operations. How to make secure NN training practical is now a hot topic in the field of cryptography. We leave this for future work.

---

> > ### Comment · Reviewer_d4vk · 2021-08-20
> > **Response**
> >
> > Thanks for the further explanation. For 1, it would be good if the authors can add a discussion section to further talk about the efficiency issue of baseline and the potential usage of the proposed method in a large-scale scenario. For 2, an analysis about potential issues or benefits brought by the application of the proposed approach to NN models will help readers to better understand the work.

---

> > > ### Author Response · Authors · 2021-08-24
> > > **Response to Reviewer d4vk**
> > >
> > > Thank you for your suggestions. We will add discussion on the efficiency issue about the baseline solutions (including the SeSoRec and secure NN protocols), and on the possibility of using our method to improve the efficiency of other secure machine learning algorithms, especially in large-scale scenarios.

---

> > > > ### Comment · Reviewer_d4vk · 2021-09-02
> > > > **Response**
> > > >
> > > > Thanks. I think this work can be an important work towards addressing the efficiency challenge in the secure recommendation scenario. I am happy to further increase my score to 7.

---

> > > ### Author Response · Authors · 2021-09-01
> > > **Thanks to Reviewer d4vk**
> > >
> > > We would like to thank you for taking the time to review our paper and finally giving positive feedback.
> > >
> > > Kindly let us know if we have adequately addressed your concerns on the lack of large-scale experiments. Note that through the discussion with reviewer Maoq, we have verified that our method works for the svd++ model, and we have also explained the potential of our proposed secure sparse matrix multiplication protocol in existing secure NN training frameworks.
> > >
> > > We truly appreciate your valuable comments that help us improve the clarity of our claim regarding how our proposed method generalizes to other recommendation models (e.g., deep learning based models).

---

### Author Response · Authors · 2021-09-11
**Further Response to Reviewer Maoq**

Dear Reviewer Maoq,

We sincerely thank you again for the suggestive comments. Based on your last comments, we have made further clarifications and provided new results on SVD++ on 02 September. We hope our last response has addressed your concerns and would appreciate your further comments.

Best,

The authors

---

### Decision · Program_Chairs · 2021-09-27

**Decision:**

Accept (Poster)

**Comment:**

The reviews were generally positive about the motivation for the problem, novelty of the techniques, and execution of the paper. They identified a few areas for potential improvement, including expanding the scope of the experiments and handling more state-of-the-art recommendation systems. These concerns were adequately addressed in author feedback. The authors convincingly explained how their techniques would generalize to other methods based on sparse matrix multiplication, and gave an analysis for SVD++ as a specific example. I recommend the paper for acceptance.

My personal enthusiasm for the paper is a touch lower than that of the reviewers because the MPC-based security guarantees of the framework hold at a per-iteration level, rather than giving an end-to-end guarantee for the whole training process. (That is, allowing each party to learn the full sequence of social updates is potentially more disclosive than allowing each party to learn the final model.) This is a weaker guarantee, and less technically interesting to achieve. Nevertheless, the security obtained does still appear to be new and meaningful.

Comment for the authors: Please precisely define the MPC-hybrid and PIR-hybrid with leakage models in which the protocols are proved secure.